# Identification of QTLs Conferring Resistance to Bacterial Diseases in Rice

**DOI:** 10.3390/plants12152853

**Published:** 2023-08-02

**Authors:** Yuan Fang, Di Ding, Yujia Gu, Qiwei Jia, Qiaolin Zheng, Qian Qian, Yuexing Wang, Yuchun Rao, Yijian Mao

**Affiliations:** 1College of Life Sciences, Zhejiang Normal University, Jinhua 321004, China; fy0579@zjnu.cn (Y.F.); l520x595l@163.com (D.D.); guyj0104@163.com (Y.G.); jiaqw@163.com (Q.J.); 2Ball Horticultural Company, West Chicago, IL 60606, USA; qzheng@ballhort.com; 3State Key Laboratory of Rice Biology and Breeding, China National Rice Research Institute, Hangzhou 310006, China; qianqian188@hotmail.com (Q.Q.); wangyuexing@caas.cn (Y.W.)

**Keywords:** bacterial panicle blight, bacterial leaf streak, bacterial brown stripe, rice, quantitative analysis

## Abstract

Bacterial panicle blight, bacterial leaf streak, and bacterial brown stripe are common bacterial diseases in rice that represent global threats to stable rice yields. In this study, we used the rice variety HZ, Nekken and their 120 RIL population as experimental materials. Phenotypes of the parents and RILs were quantitatively analyzed after inoculation with *Burkholderia glumae*, *Xanthomonas oryzae* pv. *oryzicola*, and *Acidovorax avenae* subsp. *avenae*. Genetic SNP maps were also constructed and used for QTL mapping of the quantitative traits. We located 40 QTL loci on 12 chromosomes. The analysis of disease resistance-related candidate genes in the QTL regions with high LOD value on chromosomes 1, 3, 4, and 12 revealed differential expression before and after treatment, suggesting that the identified genes mediated the variable disease resistance profiles of Huazhan and Nekken2. These results provide an important foundation for cloning bacterial-resistant QTLs of panicle blight, leaf streak, and brown stripe in rice.

## 1. Introduction

Rice (*Oryza sativa*) is one of the world’s most important food crops. Bacterial diseases of rice, particularly bacterial panicle blight of rice (BPB), also termed bacterial grain rot or seedling rot caused by *Burkholderia glumae* [1], bacterial leaf streak of rice (BLS) caused by *Xanthomonas oryzae* pv. *oryzicola* (*Xoc*) [2], and bacterial brown stripe of rice (BBS) caused by *Acidovorax avenae* subsp. *avenae* [3], are the most destructive diseases of rice that lead to a decline in rice yield and quality. Upon infection, rice leaves and other growing regions become necrotic. Due to the rapid spread of disease, current treatments fail to prevent yield losses and the destruction of farmlands [1,2,3]. BPB, BLS, and BBS have emerged as global threats to crop health. The yield reduction of rice from BPB could reach 75% in severely infested fields, resulting in grain weight reduction, florets sterility, and seed germination inhibition [4]. It has been reported that BBS leads to heavy economic losses in many rice-growing regions such as Asia, Africa, North America and Europe [5,6]. BLSs are also expanding annually, representing serious threats to global rice production and food security. It is estimated that the disease caused a 0–17% reduction in yield production [7]. In addition, many disease-resistant traits that are relevant to lesion mimic and senescence, led to a further reduction in production [8,9]. The most cost-effective method to control these bacterial diseases is the breeding of disease-resistant plants. The identification of resistance QTLs and genes on rice chromosomes are therefore important for future crop protection strategies [10,11].

To date, several BPB-resistant quantitative trait loci (QTLs) have been identified in rice. Through chromosomal hybridization between Nona Bokra (resistant) and Koshihikari (susceptible) parents, a resistant QTL *qRBS1* was detected between RM24930 and RM24944 with a 5.5 LOD score [12]. *RBG2*, another major QTL, was fine-mapped between RM1216 and RM11727 with a LOD score of 3.0 [13,14]. Pinson et al. detected 12 BPB QTLs; among them, *qBPB-3-1* is the most significant QTL explained 9.8 to 13.8% of phenotypic variance [15]. Previous studies investigated the genes governing the resistance of BLS; Tang et al. detected 11 QTLs on six chromosomes, which explained 84.6% of the genetic variation in 1997 [16]. Chen selected six SSRs of three BLS QTLs to facilitate the molecular-marker-assisted breeding and further verified the authenticity of the identified QTLs [17]. Chen et al. verified the credibility of the 11 QTLs including *qBlsr3d*, *qBlsr5a* and *qBlsr5b* using a near-isogenic line and concluded that pyramiding four or five resistant alleles in major resistant QTLs could breed a new variety of high resistance to BLS [18]. Zheng et al. identified a resistance QTL in RM279–RM154 on chromosome 2 with 13.7% phenotypic variation and 0.9576 additive effect [19]. Li and his colleagues used CSSL to fine-map *qBlsr5a* between RM5816 and RM122 with a physical distance of ~52.7 kb [20]. From the same F2 population, another QTL with a large contribution rate was detected in the RM153–RM159 on the chromosome 5 [21,22]. Cao et al. localized *qBlsr3d* between 3DSSR3 and 3DSSR12 with ~1250 kb using CSSL [23]. Ma et al. used a DH population to detect four QTLs and explained a 10.1937~19.2549 variation [24]. However, few resistant QTLs have been reported for BBS.

Although studies on these pathogenic strains have rapidly progressed and a number of resistance-related QTLs has been identified, the numbers mapped to regions of less than 1 Mbp are sparse, with few QTLs cloned and transformed due to low LOD values and/or a lack of validation in an environmental or genetic background [15,23,24]. In addition, BPB, BLS, and BBS have not been examined within a single population, and no broad-spectrum anti-bacterial disease QTLs or genes have been identified [25].

In this study, based on previous field work, a total of 120 recombinant inbred lines (RILs) of Huazhan, Nekken2 and their offspring were investigated to identify resistant QTLs of BPB, BLS, and BBS that contribute to resistance sites for the breeding of resistant rice cultivars.

## 2. Results

### 2.1. BBS Resistance of Parents and Offspring

At 3 weeks post-inoculation of the *A. avenae* subsp. *avenae* strain RS-1, a significant variation was observed in the height of parental plants. The average height of infected HZ was 104.00 cm, which is 5.89% lower than non-infected plants. The average plant height of NK2 was 101.40 cm, a decrease of 3.43%. These results reveal significant differences in BBS resistance between the two parents. At 1 week post-inoculation, a number of plants within the RIL population showed the symptoms of BBS, with brown spots appearing on the leaves and stems. At 3 weeks post-inoculation, the height of field plants varied significantly. The growth of a number of infected plants was slower than that of normal plants, and signs of decay were evident. The average height of infected plants within the RIL population was 105.40 cm, compared to 110.43 cm in non-infected plants. In addition, a number of transgressive plants also appeared and the plant height showed a normal distribution and met the requirements of QTL mapping (Figure 1A).

### 2.2. BLS Resistance of Parents and Offspring

Leaf streak symptoms in parental plants were visible and stable following inoculation of the Xoc strain into BLS256. The average length of the lesions in NK2 leaves was 1.34 cm compared to 10.22 cm in HZ leaves. These results indicated significant differences in the resistance between NK2 and HZ. The average lesion length of the RIL population was 5.87 cm, indicating a favourable crossing of the parents. It was noted that when the lesion length exceeded 10 cm, the plant coefficient decreased sharply, then maintained a continuous normal distribution thereafter. In addition, a number of transgressive plants also appeared and the plant height showed a normal distribution and met the requirements of QTL mapping (Figure 1B).

### 2.3. BPB Resistance of Parents and Offspring

Lesions formed on the uppermost sheath covering the panicles. Some grains did not fill in *B. glumae* strain LMG2196 infected plants. The rate of empty seeds in both parents increased during the heading stage. The rate of empty seeds of inoculated HZ was 0.27, compared to 0.14 in normal plants (~48% increase after inoculation). The empty seed rate of NK2 was 0.39 and 0.18 with and without bacterial infection, respectively (~54% increase after inoculation). At 1 month post-inoculation, most grains became white and empty and rice yields decreased. The average rate of empty seeds in diseased plants within the RIL population was 0.38, compared to 0.25 in untreated plants (a 34% increase after inoculation). The rate of plants with empty seeds in disease vs. normal plants was normally distributed. The peak in normal plants ranged from 0.08 to 0.16, compared to 0.40 to 0.48 in diseased plants. In addition, a number of transgressive plants also appeared and the plant height showed a normal distribution and met the requirements of QTL mapping (Figure 1C).

### 2.4. QTL Localization and Analysis

Regarding the pathogenic strains, QTLs were detected with distinguishable effect sizes at 40 loci on 12 chromosomes (Table 1, Figure 2). For BPB, HZ gave favorable alleles, and nine QTLs were detected which were distributed on chromosomes 3, 4, 5, 7, 8, 9, and 10 in rice, three of which showed larger effects. A single QTL was located between 101.81 and 102.52 cm of chromosome 3 with a threshold of 2.91 and a physical distance of ~167 kb. The other two were located on chromosome 7 between 74.35 and 77.04 cm with a threshold of 2.97 and a physical distance of ~628 kb, and between 114.37 and 123.58 cm with a 3.44 LOD score and a physical distance of ~2149 kb.

For BLS, NK2 gave favorable alleles, and 13 QTLs were detected and distributed on chromosomes 1, 3, 4, 5, 6, 7, and 12, 4 of which showed larger effects. A single QTL was located within 2.06–28.53 cm on chromosome 3 with a physical distance of ~6173 kb. A further QTL was located between 82.97 cm and 97.92 cm on chromosome 4, with a physical distance of ~3488 kb. The other two QTLs were found on chromosome 12; one was located between 0.11 cm and 21.27 cm with a threshold of up to 4.84 and a physical distance of ~494 kb, and the other within 91.33–109.31 cm with a physical distance of ~4194 kb.

For BBS, HZ gave favorable alleles, and a total of 18 QTLs was detected. Among them, qBBS-1.3 had a relatively large threshold and was located between 157.81 and 174.46 cm of chromosome 1, with a threshold of 4.33. Significantly, there were three QTLs of BBS whose intervals coincided with the other two diseases. For example, qBBS-8 and qBBS-12.2 overlapped with qBLS-8.1 and qBLS-12.3, respectively, which suggested that the overlapping regions had broad-spectrum genes against both diseases. In addition, qBBS-9 overlapped with qBPB-9, meaning the 0.4 cm region may exist micro and pleiotropic effect genes against both diseases.

### 2.5. Expression Analysis of Disease Resistance-Related Genes

We screened all functional genes in the intervals of eight large QTLs on chromosomes 1, 3, 4, 7, and 12 (http://rice.plantbiology.msu.edu, accessed on 15 August 2021) and selected candidate genes based on their function (http://ricedata.cn, accessed on 20 August 2021). The previous website showed the genes’ location so we could find all functional genes; the later website showed all selected functional genes. We then picked 27 candidate genes. Proteins encoded by these genes, including structural protein domains, transcription factors, resistance proteins, and transporters, are shown in Table 2. The expressions of each gene before and after inoculation were compared via qRT-PCR analysis (Figure 3). BLS candidate genes *LOC_Os03g10910*, *LOC_Os04g32850*, *LOC_Os12g06920*, and *LOC_Os12g09240* were significantly up-regulated in response to bacterial inoculation in NK2, while they were down-regulated in HZ. Other genes of BLS such as *LOC_Os03g10900*, *LOC_Os03g11010*, *LOC_Os03g11340*, and *LOC_Os12g39620* were down-regulated after inoculation of BLS in the parents. *LOC_Os01g68330* and *LOC_Os01g68330* were down-regulated after inoculation of BBS in the parents.

## 3. Discussion

Expression traits work cooperatively through multiple factors. QTLs of the same trait can deviate due to differences in experimental materials, environmental conditions, and treatment times. A BPB-resistant QTL *qRBS1* with an SSR marker was identified in a 393–kb interval on chromosome 10 (marker interval: RM24930–RM24944) by chromosomal fragment substitution lines derived from the hybridization of Nona Bokra (resistant) and Koshihikari (susceptible) as parents [12] (Table 3). Similarly, we located a QTL site conferring BBS-resistance on chromosome 10 (Table 1), which varied from *qRBS1* by genetic map distances, indicating the potential presence of a QTL controlling BPB resistance in rice. Moreover, the BPB QTL loci detected on chromosomes 1 and 7 were consistent with those detected previously [15,26], suggesting that both loci were indeed resistant to BPB and are applicable for resistance breeding (Table 3).

A total of 13 BLS-resistant QTLs was detected and BLS was controlled by multiple genes. Two resistant QTLs on chromosome 12 had large effects and likely represented novel sites of BLS resistance. In addition, *qBLS-12.3* overlapped with *qBBS-12.2*, with LOD scores of 4.65 and 2,26, respectively. Quantitative analysis of *LOC_Os12g06920* and *LOC_Os12g09240* localized in *qBLS-12.3* showed elevated expression in both HZ and NK2 upon bacterial infection; furthermore, more significantly in NK2, which may be responsible for a shorter NK2 lesion length. Both genes encode the anti-pathogenic protein NBS-LRR which binds to and induces conformational changes in the amino terminal and LRR domains, leading to altered kinase activity and subsequent disease-resistance responses [27]. Tang et al. localized the BLS-resistant QTL *qBlsr4b* on chromosome 4 through the analysis of RILs obtained from the highly susceptible H359 and highly resistant Acc8558 as parents [16]. Others detected four QTLs on chromosome 4 based on a double diploid population cultured from the F1 hybrid anther of Taichung local No. 1 and japonica rice Chunjiang 06 [24] (Table 3). These QTLs had overlapping intervals with the QTLs identified in this study. We therefore concluded that the major mechanism by which the QTLs control rice BLS is in this segment of chromosome 4.

Chen et al. detected the presence of BLS-resistance in the RM231–RM7 interval on chromosome 3 by examining the near-isogenic line H359R of the indica cultivar Acc8558 and H359 [18] (Table 3). This overlapped with the BLS-resistant QTLs on chromosome 3 that were identified in this study. Candidate genes within this interval including *LOC_Os03g10910* were upregulated significantly in NK2. *LOC_Os03g10910* encodes a disease-resistant protein containing the NB-ARC/LRR domain that regulates mammalian cell apoptosis and plant hypersensitivity [28]. In previous studies, the overexpression of *OsPDRH9N*, which contains a conserved NB-ARC domain, increased the sensitivity of transgenic plants to *Xoc* [28]. Thus, the overexpression of *LOC_Os03g10910* may represent a factor influencing the stronger resistance of NK2 to BLS compared to HZ.

A number of identified QTLs of BLS overlapped with the QTLs of other diseases. Fukuoka et al. cloned *Pi21*, which was mapped to a 1705–base pair (bp) region containing a single gene, *LOC_Os04g32850* [11]. *qBLS-4* (LOD = 4.67) identified in BLS in this study overlapped with this region and *Pi21* was associated with basal disease resistance. Following inoculation with toxic pathogens, pathogenicity-associated genes in plants were upregulated. Through analysis of the transcriptional levels of BLS candidate genes in this study, the expression of *LOC_Os04g32850* on chromosome 4 increased significantly in NK2 but decreased significantly in HZ upon bacterial infection. We therefore hypothesize that NK2 carries the disease-resistant allele *Pi21*, and carries the susceptible allele. Gregory et al. cloned a further rice-blast-resistant gene, *Pi-ta* [10]. Hittalmani et al. localized the gene from RG241 to RZ397 of chromosome 12, which overlapped with one of the major QTLs of BLS in this study [29]. These data suggest that the above overlapping regions contain R genes that are resistant to both BLS and Blast disease.

No BBS-resistant QTLs have been reported to date. In this study, 18 QTLs were detected, particularly chromosome 1 that had a threshold of 4.33, representing a novel QTL for BBS resistance in rice. Quantitative expression of candidate genes on chromosome 1 showed that *LOC_Os01g68330* was down-regulated in both HZ and NK2. *LOC_Os01g68330* encodes a chloroplast precursor that functions during photosynthesis in rice, the functional integrity of which is required for plant growth and development. Following BBS inoculation, the plants became shorter. In addition, *qBBS-8*, *qBBS-9* and *qBBS-12.2* were overlapped with *qBLS-8.1*, *qBPB-9* and *qBLS-12.3*, which means those regions may exist multi effect QTL locus and R genes. CSSLs can be used for fine mapping in the future. BBS causes rice decay during the seedling stage, leading to grain discoloration and rice ear withering at the earing stage. In addition, BBS is highly transmissible. BBS has catastrophic consequences with outbreaks in many countries and regions leading to serious rice losses [30,31]. To date, the disease remains in a latent state in China. When ecological and climate conditions are favorable, BBS is likely to cause an epidemic. BLS causes the yellowing and withering of rice leaves, increases the rate of empty seeds, and decreases the 1000-seed weight. This can result in a 40–60% loss of yields [32]. BBS is a seed-borne disease with an extremely high bacterium-carrying rate [33].

Antibiotics are the current treatment of choice for bacterial disease, but the emergence of drug-resistant strains and their resultant environmental pollution have seriously restricted their use. The isolation and identification of pathogenic bacteria, pathogenic mechanisms, resistance-source screening and the mining of resistant genes require further knowledge. Progress in the breeding of rice-disease-resistance plants has been slow. Here, we simultaneously investigated resistant QTLs for BPB, BLS and BBS under the same genetic background. No single resistance site for all three pathogenic diseases was identified, suggesting that the ability of rice to develop resistance to these pathogens acts through independent mechanisms. Using the near-isogenic H359R of Acc8558 and H359, Chen et al. identified *qBlsr5b* on chromosome 5 that overlapped with a BBS-resistant site identified in this study [18]. Whether this segment simultaneously regulates BPB and BLS requires further verification. We also detected candidate genes with relatively large effects within the QTL intervals and identified genes associated with disease resistance. The specific roles of these genes require further verification through genome sequencing and the identification of their interacting partners. Chromosome segment substitution lines (CSSLs) hybridized with cultivars planted in the south of China were also constructed for nine QTLs with a relatively large threshold, producing a localized ecological type with good adaptability and strong anti-bacterial-disease properties.

## 4. Materials and Methods

### 4.1. Parents and RIL Population

Rice plants were cultivated in the Experimental Stations of the China National Rice Research Institute in Hangzhou, China, during the natural growing season. The 120 RIL population was derived by single-seed descents from the cross between the *indica* rice restorer Huazhan (HZ) and a cultivar Nekken 2 (NK2). HZ is a variety with strong stress resistance, while Nekken is a widely compatible variety [25,34].

### 4.2. Linkage Map Construction and QTL Mapping

Genomic DNA were extracted from young leaves of HZ, NK2, and 120 RILs using the CTAB method [35]. Barcode multiplex sequencing libraries were constructed as per the manufacturer’s recommendations (Illumina, San Diego, CA, USA), and paired-end sequencing was performed using the Illumina X-Ten sequencer with 10× sequencing depth for 120 RILs and 50× for the RIL parents. Reads were aligned to the Nipponbare version 7 reference genome (http://rice.plantbiology.msu.edu/) using BWA-MEM version 0.7.10 [36]. SNP calling and filtration were performed using the SAMtools version 1.6 [37]. A circular ideogram displaying complete genome variation information was constructed using Circos version 0.67 [38]. Variation sites among the RILs obtained from HZ and NK2 were compared, and their genotypes determined using a hidden Markov model approach [39]. Consecutive SNP sites within the same genotype were clustered, with those less than 100 kb filtered out. R/QTL was used to locate the QTLs controlling three bacterial diseases [40]. Sequencing data were analyzed to obtain a total of 4858 markers evenly distributed on 12 chromosomes to construct the genetic maps [25].

### 4.3. Germination and Cultivation of Rice Seeds

A total of 60 seeds of HZ and Nekken and each individual plant of the RIL population were selected and immersed in water for 2 days. Following surface decanting, seeds were wrapped in moist towels and incubated at 37 °C for 48 h to accelerate germination. Seeds with a consistent germination status were selected. After 30 days, 24 plants of each line were transplanted to soil and under field conditions with an interplant spacing of 56.7 × 63.3 cm for transplanting. Regular field management was performed during this period.

### 4.4. Pathogenicity Tests of Parents and Offspring

Strains *B. glumae* LMG2196, *Xoc* BLS256 and *A. avenae* subsp. *avenae* strain RS-1 were used as pathogenic bacteria. Pathogenicity tests were performed in test fields in 2018. Test fields were located inside the campus of Zhejiang Normal University in Jinhua, Zhejiang, and separated from local farms to prevent disease transmission. Bacterial cells were grown in NB broth at 28 °C with shaking at 200 rpm for 16 h, until cells reached the exponential phase. Bacterial cells were harvested by centrifugation, washed twice, and resuspended in sterile water at an optical density at 600 nm (OD600) of 0.3 (approximately 1 × 10^8^ CFU/mL). We sprayed with the bacterial suspension of *B. glumae* when panicles were emerging (20 to 30% emerged) from the boot at the booting stage. Empty seeds were calculated by cutting four rice ears from both processed and unprocessed plants (at the heading stage) to evaluate the empty seed rate [41]. The leaves of adult plants (2 months old) were inoculated with bacterial suspensions of *Xoc*. Leaf piercing for lesion length was measured to evaluate water-soaked symptoms [42,43]. *A. avenae* subsp. *avenae* was also inoculated by injecting ~10 μL of a bacteria suspension into each stem at the late seeding stage. Plant phenotypes were monitored every 2 days after one week of inoculation. The height of infected and uninfected plants was measured 3 weeks post-inoculation [44]. Four plants were inoculated for each experiment, and each treatment was repeated at least three times.

### 4.5. QTL Localization and Analysis

Based on the high-density SNP genetic map with 4858 constructed markers, R-QTL analysis was used to map the QTL interval including the rate of plants without empty seeds after BPB inoculation, plant height after BBS infection, and the length of lesions after BLS infection. QTL analysis was performed using R/QTL [45]. LOD = 2.0 was set as the threshold value to determine the existence of the QTL.

### 4.6. Quantitative Analysis of Gene Expression

The leaves of both parents were collected for RNA extraction, with untreated parental leaves collected as controls. Total RNA was extracted from the leaves using RNA isolation kits (Invitrogen, Waltham, MA, USA). cDNA was synthesized using the ReverTra Ace^®^ qPCR RT Kit (TOYOBO, Cat No. FSQ-101, Osaka, Japan). Based on the QTL localization, genes associated with BLS and BBS resistance were selected for phenotypic analysis in the interval between chromosomes 1, 3, 4 and 12. Real-time quantitative PCR (qRT-PCR) was performed on a 7500 real-time PCR system (Applied Biosystems, Waltham, MA, USA; Life Technologies, Carlsbad, CA, USA) to assess the differential expression of each gene upon pathogen infection. All experiments were performed in triplicate. qRT-PCR analysis was performed as follows: total volume 20 μL, cDNA: 2 μL, SYBR qPCR Mix (TOYOBO): 10 μL, positive and negative primers (10 μM): 0.8 μL each, ddH2O up to 20 μL, and an amplification procedure: 95 °C for 30 s followed by 40 cycles of 95 °C for 5 s, 55 °C for 10 s, 72 °C for 15 s. Primer sequences are shown in Table 4. Experimental data were analyzed using Excel and SPSS19.0 software. A *t*-test was used to compare treatment groups.

## Figures and Tables

**Figure 1 plants-12-02853-f001:**
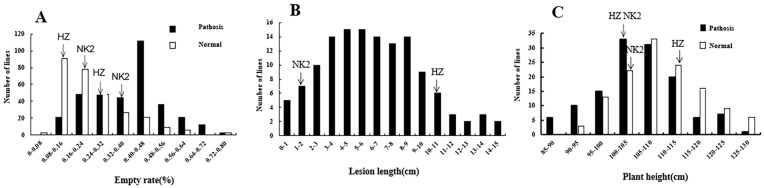
Resistance distribution in the RIL population to BPB (**A**), BLS (**B**) and BBS (**C**).

**Figure 2 plants-12-02853-f002:**
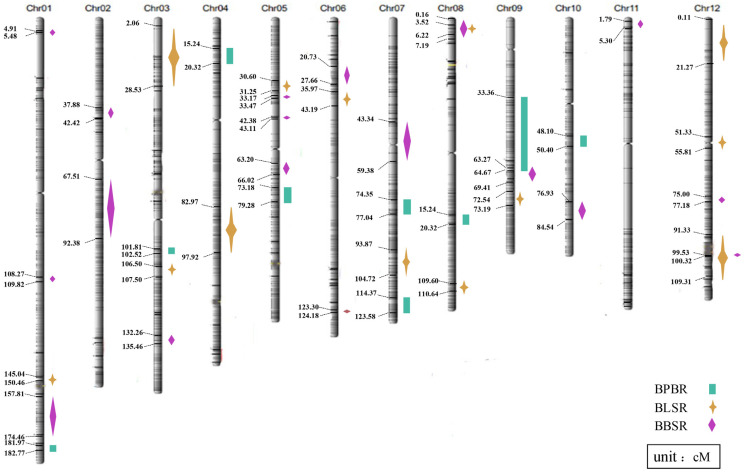
Location of QTLs for resistance to rice diseases in the rice RIL population.

**Figure 3 plants-12-02853-f003:**
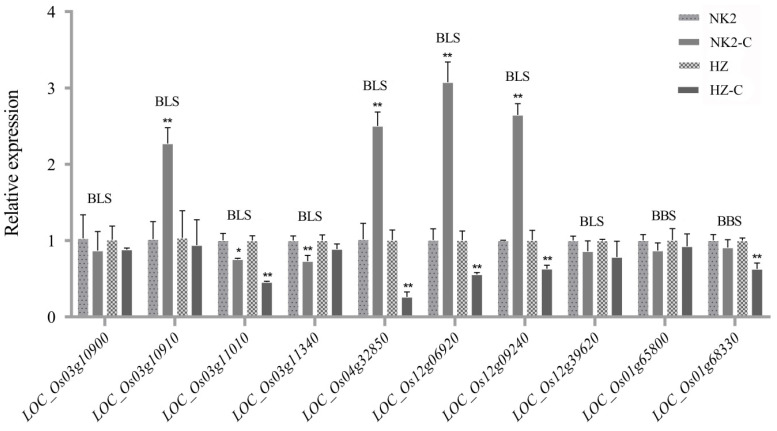
Expression of genes resistant to BBS and BLS. NK2-C and HZ-C represent bacterial treatment (BBS or BLS). * represents the significant difference at the *p* < 0.05 level; ** represents the significant difference at the *p* < 0.01 level.

**Table 1 plants-12-02853-t001:** QTL interval statistics of resistance to rice diseases in the rice RIL population.

QTL Name	Chromosome	Physical Distance	Genetic Interval	LOD Score	QTL Name	Chromosome	Physical Distance	Genetic Interval	LOD Score
*qBPB-1*	1	42,450,343–42,636,696	181.97–182.77	2.02	*qBLS-12.2*	12	11,975,136–13,018,138	51.33–55.81	2.59
*qBPB-3*	3	23,749,047–23,915,724	101.81–102.52	2.91	*qBLS-12.3*	12	21,305,692–25,499,574	91.33–109.31	4.65
*qBPB-4*	4	3,556,084–4,739,169	15.24–20.32	2.69	*qBBS-1.1*	1	1,144,806–1,279,149	4.91–5.48	2.70
*qBPB-5*	5	17,071,562–18,494,867	73.18–79.28	2.73	*qBBS-1.2*	1	25,257,953–25,619,150	108.27–109.82	2.47
*qBPB-7.1*	7	17,343,649–17,971,153	74.35–77.04	2.97	*qBBS-1.3*	1	36,814,552–40,697,439	157.81–174.46	4.33
*qBPB-7.2*	7	26,679,743–28,828,922	114.37–123.58	3.44	*qBBS-2.1*	2	8,837,300–9,896,262	37.88–42.42	3.67
*qBPB-8*	8	7,782,000–15,039,076	91.69–92.96	2.16	*qBBS-2.2*	2	15,749,688–21,552,527	67.51–92.38	2.78
*qBPB-9*	9	11,221,823–11,673,523	33.36–64.47	2.88	*qBBS-2.3*	3	30,854,578–31,600,900	132.26–135.46	2.78
*qBPB-10*	10	21,388,530–21,685,143	48.10–50.04	2.08	*qBBS-5.1*	5	7,739,685–7,809,379	33.17–33.47	2.83
*qBLS-1*	1	38,012,068–40,318,380	145.04–150.46	2.61	*qBBS-5.2*	5	9,887,145–10,057,827	42.38–43.11	2.88
*qBLS-3.1*	3	481,634–6,654,517	2.06–28.53	4.24	*qBBS-5.3*	5	14,743,764–15,400,737	63.20–66.02	2.28
*qBLS-3.2*	3	24,844,011–25,165,251	106.50–107.50	2.77	*qBBS-6.1*	6	4,835,127–6,452,921	20.73–27.66	2.84
*qBLS-4*	4	19,355,336–22,843,728	82.97–97.92	4.67	*qBBS-6.2*	6	28,764,720–28,969,988	123.30–124.18	2.21
*qBLS-5*	5	7,138,974–7,289,610	30.60–31.25	2.79	*qBBS-7*	7	10,110,871–13,852,811	43.34–59.38	2.48
*qBLS-6*	6	8,390,140–10,074,318	35.97–43.19	3.67	*qBBS-8*	8	37,535–1,676,678	0.16–7.19	2.97
*qBLS-7*	7	21,897,341–24,428,396	93.87–104.72	3.73	*qBBS-9*	9	14,761,639–16,192,828	63.27–69.41	2.24
*qBLS-8.1*	8	820,014–1,451,568	3.52–6.22	3.25	*qBBS-10*	10	17,947,130–19,722,649	76.93–84.54	2.55
*qBLS-8.2*	8	25,566,791–25,809,447	109.60–110.64	3.00	*qBBS-11*	11	419,178–1,236,632	1.79–5.30	2.58
*qBLS-9*	9	16,921,413–17,073,957	72.54–73.19	2.68	*qBBS-12.1*	12	17,494,749–18,004,877	75.00–77.18	2.59
*qBLS-12.1*	12	26,231–4,961,252	0.11–21.27	4.84	*qBBS-12.2*	12	23,219,979–23,404,674	99.53–100.32	2.26

**Table 2 plants-12-02853-t002:** Analysis of candidate genes in the detected QTL intervals.

	BPB		BBS		BLS
Gene ID	Putative Function	Gene ID	Putative Function	Gene ID	Putative Function
*LOC_Os02g18000*	disease resistance protein RGA2	*LOC_Os01g65800*	powdery mildew resistant protein 5, putative, expressed	*LOC_Os03g10900*	disease resistance protein
*LOC_Os07g29600*	zinc finger, C3HC4 type, domain containing protein	*LOC_Os01g68330*	antigen peptide transporter-like 1, chloroplast precursor	*LOC_Os03g10910*	NB-ARC/LRR disease resistance protein
*LOC_Os07g29820*	NBS-LRR disease resistance protein	*LOC_Os01g68630*	leaf senescence related protein	*LOC_Os03g11010*	resistance-associated macrophage protein
*LOC_Os09g13570*	bZIP transcription factor	*LOC_Os02g32160*	methyl esterase-like gene	*LOC_Os03g11340*	leucine-rich repeat resistance protein, putative, expressed
*LOC_Os09g14010*	disease resistance protein RPS2	*LOC_Os02g32980*	Germin-like protein	*LOC_Os04g32850*	Rice blast resistance gene
*LOC_Os09g14490*	TIR-NBS type disease resistance protein	*LOC_Os05g25770*	WRKY transcription factor	*LOC_Os12g06920*	NBS-LRR disease resistance protein
*LOC_Os09g16000*	Rice blast resistance gene	*LOC_Os06g10660*	Lysin motif-containing proteins	*LOC_Os12g09240*	NBS-LRR disease resistance protein
*LOC_Os09g21796*	OsFBX327—F-box domain containing protein	*LOC_Os09g25060*	WRKY transcription factor	*LOC_Os12g39620*	disease resistance protein, putative, expressed
*LOC_Os09g25060*	WRKY transcription factor	*LOC_Os09g25070*	WRKY transcription factor	*LOC_Os12g40570*	WRKY transcription factor

**Table 3 plants-12-02853-t003:** Resistant QTL reported in the candidate regions.

Disease	Chromosome	Materials	Marker Interval	Phenotypic Variance (%)	Reference
BPB	1	RILs developed from the cross of Lemont and TeQing	RG236-C112x	3.6	[15]
	7	RILs developed from the cross of Lemont and TeQing	BCD855-CDO497	2.8	[15]
	10	44 CSSLs developed from the cross of Nona Bokra and Koshihikari	RM474-RM7361	22	[12]
BLS	3	The near-isogenic line H359R developed from the cross of H359R Acc8558 and H359	RM231–RM7	/	[18]
	4	RILs developed from the cross of H359 and Acc8558	C624-C246	/	[16]
	4	DH developed from the cross of TN1 and CJ06	RM252-RM241	10.1937	[24]

**Table 4 plants-12-02853-t004:** Primer sequence of quantitative real-time fluorescent PCR.

Gene ID	Forward Primer(5′-3′)	Reverse Primer(5′-3′)
*OsActin1*	TGGCATCTCTCAGCACATTCC	TGCACAATGGATGGGTCAGA
*LOC_Os01g65800*	CCGGAAATGGGAGAATGCTG	GCAAACCTCCAGCCTCAAAA
*LOC_Os01g68330*	GTTGGACAGGAACCTAGGCT	AGCCCACTCCACTTCTTCAT
*LOC_Os03g10900*	TCGGCCTCAGATACCTCAAC	CGTTCCTTGGTATGCTGTCG
*LOC_Os03g10910*	GCGTCCGATCTTTGAAGTCC	CTTGTTCGGGTGGTCATGG
*LOC_Os03g11010*	TCTATGCCACTGAAGTCCGG	CGGAGGCGAGAATACAGTGA
*LOC_Os03g11340*	GGTCCTTTCCCTACAGCAGT	GAATGGGCCCTGTCAACTTG
*LOC_Os04g32850*	GCGATGCCAAGATCAGGAAG	CCCTGTTGTTCTTCACGTCG
*LOC_Os12g06920*	TTGGTGCACTGGGTCAATTG	CCATCGCAGGGAGTACATCT
*LOC_Os12g09240*	ACTCACTCCTCACCAAGCTC	TCAAGCTCATCCATGCATGC
*LOC_Os12g39620*	GCGGCTCTTTCTGTCTTCTG	CGCACCGATAACCTTCAGTG

## Data Availability

Not applicable.

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
