# Peer review of "Identification of QTLs Conferring Resistance to Bacterial Diseases in Rice"

_plants, 2023, doi:10.3390/plants12152853_

Round 1

Reviewer 1 Report

see my suggestions in attached pdf

Author Response

Response to Reviewer #1:

  1. L78: how you can say significant variation, was you applied any statistical test

Response: Thank you for your suggestions. We used t-test to test for significant differences in data.

  1. where is fig 1

Response: Sorry for the typo. We have corrected this in the new version.

  1. Table 1 add R squre values and additive effect values

Response: Thank you for your suggestions, since we used R/QTL method to locate the QTLs, This method didn’t contain R squre values and additive effect values, and LOD score was also an credible index to estimate whether the QTL was a major locus controlling quantitative traits (Arends et al., 2010; Ren et al., 2016). In our research, we detected several QTLs which LOD score were above 4, indicated they were credible major effect site.

  1. lod score in table is compromised generally we keep 3 LOD but here most of the qtl approvedbelow 3LOD

Response: Good suggestions, we set LOD of 2.0 as threshold of effective QTL. Because we did’t want to let go of minor QTLs, some minor QTLs can also effect traits and we wanted to show all QTLs that related to traits. So we put them in the manuscript. But we mainly focused and found candidate genes in major QTLs which had high LOD scores. The candidate genes were in major QTLs, qRT-PCR showed their expression levels. What were mainly discussed in our manuscript.

  1. Table 1 which parent gave favorable alleles for each QTL

Response: Good suggestions, as shown in Figure 1, for empty rate QTL, HZ gave favorable alleles. For lesion length QTL, NK2 gave favorable alleles. For plant height QTL, HZ gave favorable alleles.

  1. which language in box

Response: Sorry for the typo. We have corrected this in the new version.

  1. In which year cross was made, on what basis these aprents were selected?In my opinion, population size is very small, result on the based of such population may be over estimated Consider two SNPs in the coding region of glycoprote in A and glycoprotein B that change the amino acid sequence. Both of the proteins are on chromosome 4 and are found on the outside of red blood cells.

Response: Thank you for your suggestions. Through multiple generations of self pollination, each strain in the RIL population is homozygous, and the method of Linkage map construction and QTL mapping was the same of the mwthod of “A strigolactone biosynthesis gene contributed to the green revolution in rice” on Mol Plant, we used the same method and 120 RIL population, that article provides detailed information about the group, and we have also cited it in this article.

  1. In which design and how many replications experiment was laid in field, row to row and plant to plant spacing?

Response: Good suggestions. The field plant spacing should be at least 0.5m or more for avoid interference, and the experiment was repeated three times.

  1. rewrite L283-284

Response: Good suggestions. We have rewrite L283-284.

Reviewer 2 Report

the authors need to improve introduction part and make deep discussion

Just minor revision 

Author Response

Thank you for your affirmation.  We have improved the introduction part and made deep discussion in new version.

Reviewer 3 Report

The peer-reviewed paper presents a well-documented study that used an appropriately designed methodology. The overall arrangement of the manuscript is in accordance with the requirements of the editorial board of the publisher; however, to improve the quality of the presentation of results, I propose to make some minor corrections:

1 - please avoid to use the self pronounsiations  (we, our, ...etc).

2- please  add new references that support the introduction and emphasize the idea.

3- Table 1 had no footnotes that explain the appriviations, please do that.

4- the first figure was Figure 2;  where is figure 1 ????

5- in Fig 2 ,the resolution should be improved.

6- It may suitable to provide Table 2 as a landscape page view to be more clear.

7- please correct figure 3 to be figure 2 , or add figure 1 if missed?

8- please rewrite the disscusion to be more supporting the results.

9- could you please provide the recommendation by the end of  disscusion or add the results conclosion.

10 - please make sure to improve the abstract to be reflected by the pioneer results.

Minor correction are need to improve the quality of the manuscript

Author Response

Response to Reviewer #3:

  1. Please avoid to use the self pronounsiations  (we, our, ...etc).

Response:Thank you for your suggestion. We have remove the self pronounsiations in new edition.

  1. Table 1 had no footnotes that explain the appriviations, please do that.

Response: Thank you for your suggestion. Since this QTL is discovered through R/QTL software, we have explained the source and how we found it in the article.

  1. The first figure was Figure 2;  where is figure 1 ????

Response: Sorry for the typo. We have corrected this in the new version.

  1. In Fig 2 ,the resolution should be improved.

Response: Thank you for your suggestion. We have improved the resolution in the new version.

  1. It may suitable to provide Table 2 as a landscape page view to be more clear.

Response: Good suggestion. We have made Table 2 as a landscape page.

  1. Please correct figure 3 to be figure 2 , or add figure 1 if missed?

Response: Sorry for the typo. We have corrected this in the new version.

Round 2

Reviewer 1 Report

not not revised as no improvement was found except figure added, revise as per comments given on original version again attaching here

Author Response

Reviewer #1:

  1. L78: how you can say significant variation, was you applied any statistical test

Response: Thank you for your suggestions. We used t-test to test for significant differences in data.

  1. where is fig 1

Response: Sorry for the typo. We have corrected this in the new version.

  1. Table 1 add R squre values and additive effect values

Response: Thank you for your suggestions, since we used R/QTL method to locate the QTLs, This method didn’t contain R squre values and additive effect values, and LOD score was also an credible index to estimate whether the QTL was a major locus controlling quantitative traits (Arends et al., 2010; Ren et al., 2016). In our research, we detected several QTLs which LOD score were above 4, indicated they were credible major effect site.

  1. lod score in table is compromised generally we keep 3 LOD but here most of the qtl approvedbelow 3LOD

Response: Good suggestions, we set LOD of 2.0 as threshold of effective QTL. Because we did’t want to let go of minor QTLs, some minor QTLs can also effect traits and we wanted to show all QTLs that related to traits. So we put them in the manuscript. But we mainly focused and found candidate genes in major QTLs which had high LOD scores. The candidate genes were in major QTLs, qRT-PCR showed their expression levels. What were mainly discussed in our manuscript.

  1. Table 1 which parent gave favorable alleles for each QTL

Response: Good suggestions, as shown in Figure 1, for empty rate QTL, HZ gave favorable alleles. For lesion length QTL, NK2 gave favorable alleles. For plant height QTL, HZ gave favorable alleles.

  1. which language in box

Response: Sorry for the typo. We have corrected this in the new version.

  1. In which year cross was made, on what basis these aprents were selected?In my opinion, population size is very small, result on the based of such population may be over estimated Consider two SNPs in the coding region of glycoprote in A and glycoprotein B that change the amino acid sequence. Both of the proteins are on chromosome 4 and are found on the outside of red blood cells.

Response: Thank you for your suggestions. Through multiple generations of self pollination, each strain in the RIL population is homozygous, and the method of Linkage map construction and QTL mapping was the same of the mwthod of “A strigolactone biosynthesis gene contributed to the green revolution in rice” on Mol Plant, we used the same method and 120 RIL population, that article provides detailed information about the group, and we have also cited it in this article.

  1. In which design and how many replications experiment was laid in field, row to row and plant to plant spacing?

Response: Good suggestions. The field plant spacing should be at least 0.5m or more for avoid interference, and the experiment was repeated three times.

  1. rewrite L283-284

Response: Good suggestions. We have rewrite L283-284.

Round 3

Reviewer 1 Report

I am not able to see resolution of queries, Please provide rebuttal file and upload track change file to evaluate the response to suggestion. again attaching file with queries.

Author Response

Dear Editor,

   We appreciate the comments and suggestions from you and the reviewer1 on our manuscript submitted last week (Tracking#: plants-2465565); these suggestions help us to improve our manuscript substantially. The decision rendered is resubmit again last time. Thank you for re-considering the resubmitted manuscript entitled " Identification of QTLs conferring resistance to bacterial diseases in rice " to be published in Plants.

In the revised manuscript, we addressed all of the reviewers1’ concerns by providing additional explanation, analyses, including more details, and adding more validation. Here, we briefly summarize as follows:

Reviewer #1:

L78: how you can say significant variation, was you applied any statistical test

Response: Thank you for your suggestions. We used t-test to test for significant differences in data.

where is fig 1

Response: Sorry for the typo. We have corrected this in the new version.

Table 1 add R squre values and additive effect values

Response: Thank you for your suggestions, since we used R/QTL method to locate the QTLs, This method didn’t contain R squre values and additive effect values, and LOD score was also an credible index to estimate whether the QTL was a major locus controlling quantitative traits (Arends et al., 2010; Ren et al., 2016). In our research, we detected several QTLs which LOD score were above 4, indicated they were credible major effect site.

lod score in table is compromised generally we keep 3 LOD but here most of the qtl approved below 3LOD

Response: Good suggestions, we set LOD of 2.0 as threshold of effective QTL. Because we did’t want to let go of minor QTLs, some minor QTLs can also effect traits and we wanted to show all QTLs that related to traits. So we put them in the manuscript. But we mainly focused and found candidate genes in major QTLs which had high LOD scores. The candidate genes were in major QTLs, qRT-PCR showed their expression levels. What were mainly discussed in our manuscript.

Table 1 which parent gave favorable alleles for each QTL

Response: Good suggestions, as shown in Figure 1, for empty rate QTL, HZ gave favorable alleles. For lesion length QTL, NK2 gave favorable alleles. For plant height QTL, HZ gave favorable alleles.

which language in box

Response: Sorry for the typo. We have corrected this in the new version.

In which year cross was made, on what basis these aprents were selected?

Response: Good suggestions, we have revised the MS, HZ is a variety with strong stress resistance, while Nekken is a widely compatible variety. And the parents were the same of the method of “A strigolactone biosynthesis gene contributed to the green revolution in rice” on Mol Plant, we have already cited it in the article.

In my opinion, population size is very small, result on the based of such population may be over estimated Consider two SNPs in the coding region of glycoprote in A and glycoprotein B that change the amino acid sequence. Both of the proteins are on chromosome 4 and are found on the outside of red blood cells.

Response: Thank you for your suggestions. Through multiple generations of self pollination, each strain in the RIL population is homozygous, and the method of Linkage map construction and QTL mapping was the same of the method of “A strigolactone biosynthesis gene contributed to the green revolution in rice” on Mol Plant, we used the same method and 120 RIL population, that article provides detailed information about the group, and we have also cited it in this article.

Which parameters were used during R/QTL run

Response: Thank you very much for your questions. R/qtl is an extensible, interactive environment for mapping quantitative trait loci (QTL) in experimental crosses. It is distributed as source code for unix or compiled code for Windows or Mac.  We can find the parameters which were used during R/QTL run from the website of https://rqtl.org/.

In which design and how many replications experiment was laid in field, row to row and plant to plant spacing?

Response: Good suggestions. The field plant spacing should be at least 0.5m or more for avoid interference, we have revised the MS. and the experiment was repeated three times.

rewrite L283-284

Response: Good suggestions. We have rewrite L283-284.

Finally, we would like to thank you again for all helpful comments, and hope that the new version of the manuscript is suitable for publication. We look forward to your response.

Yours faithfully,

Yuan Fang, PhD

College of Life Sciences, Zhejiang Normal University, Jinhua 321004, China;

Email: fy0579@zjnu.cn
